# Priming Watermelon Resistance by Activating Physiological Response and Defense Gene Expression to Alleviate Fusarium Wilt in Wheat-Watermelon Intercropping

Huifang Lv [1,2,3], Junyang Lu [2], Yuan Huang [2], Mingxia Wang [3], Congsheng Yan [3] and Zhilong Bie [2,*]

1   School of Life Sciences, Hefei Normal University, Hefei 230601, China
2   Key Laboratory of Horticultural Plant Biology, Ministry of Education, College of Horticulture and Forestry Sciences, Huazhong Agricultural University, Wuhan 430070, China
3   Horticultural Research Institute, Anhui Academy of Agricultural Sciences, Hefei 230001, China
*   Correspondence: biezl@mail.hzau.edu.cn; Tel./Fax: +86-27-87282010

**Abstract:** Wheat intercropping in watermelon could provide relief from the occurrence of Fusarium wilting of watermelon, a severe soil-borne disease caused by the fungus *Fusarium oxysporum* f. sp. *niveum* (FON). The current study aims to investigate the effect of root exudates from three wheat cultivars and one watermelon cultivar on the growth of FON and the responses of Fusarium wilt in watermelon to intercropping with wheat. The results revealed the contrasting effects of root exudates on the mycelial growth of FON; the wheat root exudates inhibited the mycelial growth of FON, and watermelon root exudates promoted the mycelial growth of FON. Watermelon plants suffered less Fusarium wilt in the intercropping system than in the monocropping system. Wheat intercropping reduced the incidence of Fusarium wilt in watermelon, and this effect was associated with the role of wheat root exudates that inhibited the growth of FON. Malondialdehyde (MDA) contents decreased in the intercropping system compared with the monocropping system after FON inoculation. The catalase (CAT), superoxide dismutase (SOD), ascorbate peroxidase (APX), and polyphenol oxidase (PPO) activities, and total phenolics and flavonoid contents in the roots of watermelon in the intercropping system were significantly higher than those in the monocropping system. Real-time PCR analysis showed that *ClCAT*, *ClSOD*, *ClAPX*, and *ClPPO* defensive enzymes and *ClPDF2.1* and *ClPDF2.4* defensin-like gene expression were significantly induced during the early stage after FON inoculation in the intercropping system compared to the monocropping system, while peroxidases did not show a significant response to FON infection. It is suggested that intercropping with wheat alleviates Fusarium wilt of watermelon by reducing the population of FON in rhizospheric soil and activating physiological responses and defense gene expression to protect watermelon from FON infection and improve the resistance of watermelon to FON in the intercropping system.

**Keywords:** wheat intercropping; watermelon; root exudates; *Fusarium* wilt; defensive enzyme; gene expression

## 1. Introduction

Watermelon is an important fruit and is commercially cultivated. Fusarium wilt is one of the most severe soil-borne diseases caused by *Fusarium oxysporum* f. sp. *niveum* (FON). The intercropping of wheat and watermelon systems enhanced crop diversity and provided resistance against biotic stresses. Multiple studies have explained that intercropping improves plant growth [1,2] and suppresses soil-borne diseases [3]. For example, maize/pepper intercropping can effectively inhibit pepper blight in the field [4]; watermelon with aerobic rice intercropping checked the Fusarium wilt of watermelon [1]; and maize/soybean intercropping inhibited the incidence of soybean red crown rot [5].

Plant root exudates indirectly play an essential role in soil health by providing pathogens with a substrate supply that could lead to quick colonization and infection

by them in the continuous monocropping system [6]. The amount and superiority of root exudates are generally susceptible to the effects of different plants, growth environments, and plant developmental factors. A slight variation in root exudate compounds may cause significant alterations on the microbial community of soil rhizospheres [7,8]. The root exudate compositions differed between different plants, which produced different allelochemicals. In plant-soil pathogens feedback, some allelochemicals released from many plants' root exudates inhibit soil-borne pathogens' growth, while others promote [9,10]. Xu et al. [11] demonstrated that the FON mycelial growth was affected by adding wheat root exudates.

The malondialdehyde (MDA) level is commonly known as a marker of membranous lipid peroxidation [12] and is usually increased under stress conditions caused by pathogen infection [13]. Apart from MDA content, other stress-related enzymes such as catalase (CAT), superoxide dismutase (SOD), peroxidase (POD), ascorbate peroxidase (APX), and polyphenol oxidase (PPO) were also increased under stress conditions or when the plants were subjected to stress [14–16]. *Trichoderma harzianum* T23 boosts the activities of several antioxidant enzymes, for example, PPO, POD, and SOD, improving the resistance of eggplants to Fusarium wilt [17]. The SOD activity of leaves was significantly enhanced in tomato seedlings after infection by root-knot nematodes [18]. Ascorbate peroxidase (APX) is the most active oxygen species (AOS)-scavenging enzyme, which influences the metabolism of hydrogen peroxide ($H_2O_2$) in higher plants. The endogenous APX effectively maintains the antioxidant that provides resistance against oxidative damage from biotic and abiotic stresses in plants [19]. In soybean, POD, SOD, and CAT activities in leaves were increased after *Trichoderma harzianum* T-aloe pretreatment and in response to *Sclerotinia sclerotiorum*. At the same time, the $H_2O_2$ and the superoxide radical ($O_2^-$) decreased [20]. Together, these changes were attributed to gene regulation.

Our preliminary experiments found that wheat intercropping controlled Fusarium wilt in watermelon. However, there is limited data on the interaction and relationship between wheat-watermelon intercropping and FON in watermelon. To address this question, we investigated the influence of wheat and watermelon root exudates on FON mycelial growth and the effects of intercropping wheat on watermelon Fusarium wilt disease. Herein, our hypothesis was that wheat intercropping improved the resistance of watermelon to FON by regulating enzymatic activities, flavonoid and total soluble phenolic contents, and the expressions of defensive enzyme genes and defensin genes in watermelon seedlings. These data provide a mechanism underlying the interactions of wheat-watermelon intercropping and FON in watermelon.

## 2. Materials and Methods

### 2.1. Plant Materials and Experimental Design

The experiment was performed in a greenhouse at Huazhong Agricultural University (Wuhan, China). The seeds of watermelon (*Citrullus lanatus*) Variety 8424 used in the current study were obtained from the Research Center of Hami-melon, Xinjiang Academy of Agricultural Sciences. Wheat cultivars Emai 352 (E352), Emai 18 (E18), and Zhengmai 9023 (Z9023) were obtained from the Research Institute of Grain Crop, Hubei Academy of Agricultural Sciences. The *Fusarium oxysporum* f. sp. *niveum* strain was isolated from a symptomatic watermelon plant. Experiments were performed on three biological replicates. In the first experiment, root exudates were collected and treated according to the methods described in [11] and [21]. The seedlings of watermelon and wheat were maintained (28 °C/16 °C day/night) in a phytotron for 30 days. When collecting root exudates, the plants of watermelon and wheat were washed with distilled water, placed in cups containing deionized water for 48 h, and filtered with 0.45 μm Millipore membranes. For producing the concentrated extract (0.5 mL per plant$^{-1}$) the root exudates were diluted to four concentrations for analysis of FON mycelial growth. The root exudates of watermelon and wheat were assayed in petri dishes (90 mm diameter) at concentrations of 0.5 mL plant$^{-1}$, 1 mL plant$^{-1}$, 5 mL plant$^{-1}$, and 10 mL plant$^{-1}$. In each petri dish, 19 mL of

PDA medium was mixed with 1 mL of watermelon or wheat root exudates. The root points (5 mm) with fungus were taken from the edges and placed in the petri dishes. Controls contained 1 mL of sterile distilled water added to the PDA medium instead of root exudate. Then the petri dishes were incubated at 28 °C under a dark environment for 3 days, and the growth of the colonies was determined from two different directions. In the second experiment, the assay included the following two cropping systems. One is watermelon plants grown in a monoculture system (M). The other one is watermelon with E18 wheat grown in the intercropping system (I). Watermelon plants at the four-leaf stage were transplanted carefully into plastic pots with a 34 cm diameter and 24 cm height, each containing 8 L of a moist inoculated mixture of peat and vermiculite ($v/v$, 1:1). Per pot, soil contained 8 mL of $2.0 \times 10^6$ CFU mL$^{-1}$ FON spore suspension. The rhizosphere soils were sampled carefully, mixing the media adhered to the roots. The soil samples were collected and preserved at $-80$ °C to quantify the FON population. The dry weight of whole plants, wilt incidence, and the number of FON analyses were conducted 15, 25, and 35 days after transplanting. In the third experiment, the assay included three treatments: (1) watermelon plants grown in a monoculture system (M); (2) watermelon plants grown in a monoculture system with FON inoculation; and (3) watermelon with E18 wheat grown in the intercropping system with FON inoculation (IF). Alike experiment 2, plants at the four-leaf stage were transplanted into the pots. After seven days, plants in MF and IF were treated with a conidia suspension of FON ($2.0 \times 10^6$ mL$^{-1}$; 100 mL per pot). Meanwhile, the Hoagland nutrient solution was provided to meet the nutritional needs of the plants during the experiments [22]. Root samples were harvested from monocropping and watermelon with wheat intercropping watermelon plants after FON inoculation at the indicated times, frozen immediately in liquid nitrogen, and stored at $-80$ °C before analyses. Subsequently, enzyme activity, flavonoid and total phenolic contents, MDA and H$_2$O$_2$ contents, and gene expression analyses were conducted.

### 2.2. Plant Growth Analysis

Samples were harvested 15, 25, and 35 days after transplanting and used to measure the dry weight of plants and root morphology. Root scanning was performed using Imagery Scan Screen (Epson Expression 10000XL, Regent Instruments, Québec, QC, Canada). The samples were placed in an oven at 85 °C for drying and then weighed.

### 2.3. Assessment of Disease Incidence

The percentage of disease incidence was assessed as described by Wu et al. [23]. Disease incidence was defined as the percentage of infected disease plants at 15, 25, and 35 days after transplanting, respectively.

### 2.4. Soil DNA Extraction and qPCR Amplification

Total genomic DNA was isolated from rhizosphere soil using a PowerSoil DNA isolation kit (Omega Bio-Tek, Inc., Norcross, GA, USA). The populations of FON were quantified by quantitative real-time PCR (qRT-PCR). The primers used to determine FON were FON-1/ FON-2 [24]. qPCR amplification reactions were made as described by Lv et al. [3]. The standard curves were generated following previous research by Zhou et al. [25]. The populations of FON were expressed as previously described by Wakelin et al. [26].

### 2.5. Total RNA Extraction and Gene Expression Analysis

The RNA was extracted in triplicate from the roots of each treatment by using TransZol reagent (TransGen Biotech, Inc., Beijing, China) per the manufacturer's instructions. The cDNA template for qRT-PCR was synthesized from 1 μg of total RNA using HiScriptII QRT SuperMix. qRT-PCR analysis was carried out after FON inoculation, according to a previous report [27]. Primes *ClCAC* and *ClTUA* were used as the internal controls [28]. Primers for the watermelon genes *ClPDF2.1*, *ClPDF2.4*, *CHI*, *APX*, and *PPO* were designed as previously described [29], and those for other genes were generated from the gene CDS

(coding DNA sequences) [Watermelon (97103) Genome (v1)] in the Cucurbit Genomics Database 1. The primers used for the gene amplification are listed in Table 1. The gene expression levels were calculated while following the methods developed by Livak and Schmittgen [30].

**Table 1.** Gene-specific primers designed for qRT-PCR analysis.

| Gene | Forward Primer | Reverse Primer |
|---|---|---|
| *ClCAC* | 5′-AATTGTGGTTGATGCTGCAC-3′ | 5′-TGACAGCTGTACCTGGCATC-3′ |
| *ClTUA* | 5′-CTTGCTGGGAGCTCTATTGC-3′ | 5′-AACGGATTAAAAGCGTCGTG-3′ |
| *ClAPX* | 5′-CACCCTGGTAGAGAGGACAAAC-3′ | 5′-TCAAAGATGAGAGGGTTGGAAG-3′ |
| *ClPPO* | 5′-GCAAGAAGGAGAAAGAGGATGA-3′ | 5′-CTTCAAGCAATTCGGTTATTCC-3′ |
| *ClPOD* | 5′-AGTGGGTGGGTTGACCTTTCT-3′ | 5′-ATCACAAAGGGCTTCTCCAAA-3′ |
| *ClCAT* | 5′-GCTCACCATGCCGAGAGGTATC-3′ | 5′-CGTTCTTGCCTGTCTGATGTCC-3′ |
| *ClSOD* | 5′-TCTCAAGCTACCTCGCCACTC-3′ | 5′-AGCGTGACGACGCCTTCAAC-3′ |
| *ClCHI* | 5′-CTGAATTCTTGGAGTCAGTGGA-3′ | 5′-ACGCCTTGCTCCATAACATAAC-3′ |
| *ClPDF2.1* | 5′-ATGAAGTTCTTTTCCGCTGC-3′ | 5′-TCAAACGCAGTGCTTTGTGCAGAAG-3′ |
| *ClPDF2.4* | 5′-ATGAAGTTTCTTTTTCAGCTGC-3′ | 5′-TCAAACGCAGTGCTTTGTG-3′ |

Note: *ClCAC* and *ClTUA* were used as reference genes.

### 2.6. Defensive Enzyme Assays

The defensive enzyme was measured as previously described [27]. After centrifuging at $12,000\times g$ for 20 min at 4 °C, the supernatants were gently transferred to sterilized tubes to measure the enzyme activity. CAT activity was determined by a decrease in $A_{240,}$ as described by Patra et al. [31]. We determined the SOD activity by monitoring the change in $A_{560}$ as described by Dhindsa et al. [32]. APX activity was calculated as a decline in $A_{290}$ per the method Nakano and Asada described [33]. PPO activity was determined as an increase in $A_{398}$ by using catechol as a substrate [34]. Subsequently, the POD activity was determined by examining the increase in $A_{470}$ nm by using guaiacol as a substrate [35].

### 2.7. Determination of Total Phenolic and Flavonoid Contents

Root tissues were crushed in liquid nitrogen with a pestle and mortar until a fine powder. Thus, 0.1 g of the sample was transferred to a frozen tube in liquid nitrogen, homogenized with 1.5 mL of 80% methanol, and extracted overnight in a rotary shaker (150 rpm) at room temperature. The homogenized solutions were centrifuged, and the supernatants were used to determine the phenolic and flavonoid contents. Total phenolics were determined following the method explained by Arnaldos et al. [36] and Ruiz et al. [37]. Flavonoids were determined as described by Tekel'ova et al. [38].

### 2.8. Determination of MDA and $H_2O_2$ Contents

Heath and Packer's [39] methods were utilized for MDA content assays. Then, 0.3 g root tissues were ground to homogenize with 5% TCA 5 mL. The homogenates were centrifuged at 4 °C for 20 min at $4000\times g$, and the supernatants were used to determine the MDA content.

On the other hand, 0.2 g of root tissues were extracted in 3 mL of 1 M $HClO_4$ to measure $H_2O_2$ content. The extracts were centrifuged at $12,000\times g$ for 10 min at 4 °C, and the resulting supernatants were filtered through a Sep-Pak C18 cartridge adjusted to pH 7.0. $H_2O_2$ contents were measured according to the method described in [40].

### 2.9. Statistical Analysis

The bioassays were analyzed using SPSS 19.0 analysis software (IBM Corporation, New York, NY, USA). An independent sample t-test was used to evaluate significant

differences between treatments at $p < 0.05$. Tukey's test was used to evaluate significant differences among treatments at $p < 0.05$. All data were expressed as mean $\pm$ standard error.

## 3. Results

### 3.1. Effects of Wheat and Watermelon Root Exudates on Mycelial Growth of FON

The FON mycelial growth analysis revealed that watermelon root exudates significantly promoted the mycelial growth of FON after 3 days of treatment compared with controls. On the contrary, E18, E352, and Z9023 wheat root exudates resist the mycelial development of FON. E18 wheat root exudates have a greater inhibitory effect compared with E352 and Z9023, and the peaked resistance rate was 9% at 5 mL plant$^{-1}$ (Table 2).

**Table 2.** Effects of three wheat and one watermelon root exudates on mycelial growth of FON.

| Cultivars | Colony Growth Diameters of FON (cm) | | | |
|---|---|---|---|---|
| | 0.5 mL Plant$^{-1}$ | 1 mL Plant$^{-1}$ | 5 mL Plant$^{-1}$ | 10 mL Plant$^{-1}$ |
| Control | 3.34 $\pm$ 0.044 b | 3.34 $\pm$ 0.044 b | 3.34 $\pm$ 0.044 b | 3.34 $\pm$ 0.044 b |
| E18 | 3.02 $\pm$ 0.026 c | 3.04 $\pm$ 0.018 d | 3.04 $\pm$ 0.009 d | 3.12 $\pm$ 0.040 c |
| E352 | 3.06 $\pm$ 0.029 c | 3.10 $\pm$ 0.004 cd | 3.14 $\pm$ 0.041 c | 3.16 $\pm$ 0.027 c |
| Z9023 | 3.09 $\pm$ 0.016 c | 3.15 $\pm$ 0.015 c | 3.18 $\pm$ 0.017 c | 3.18 $\pm$ 0.010 c |
| Watermelon | 3.63 $\pm$ 0.012 a | 3.53 $\pm$ 0.039 a | 3.46 $\pm$ 0.031 a | 3.44 $\pm$ 0.018 a |

All values are mean $\pm$ SE (n = 3). The different letters in the same concentrations indicate statistically significant differences among treatments of mean values ($p < 0.05$, Tukey's test).

### 3.2. Effect of Wheat Intercropping on Plant Growth of Watermelon

In the intercropping system, the watermelon plants' root length, surface area, root volume, root numbers, and root dry weight were higher than in the monocropping system 35 days after transplanting (Table 3).

**Table 3.** Effects of watermelon monocropping and wheat intercropping on root morphology in watermelon.

| Treatment | Days | Length (cm) | SurfArea (cm$^2$) | Volume (cm$^3$) | AvgDiam (mm) | Number | Dry Weight (g) |
|---|---|---|---|---|---|---|---|
| M | 15 | 863.14 $\pm$ 40.25 a | 123.49 $\pm$ 1.75 a | 1.43 $\pm$ 0.06 a | 0.36 $\pm$ 0.02 a | 4916.00 $\pm$ 55.64 a | 0.17 $\pm$ 0.01 a |
| I | | 977.77 $\pm$ 13.77 a | 154.02 $\pm$ 12.24 a | 1.68 $\pm$ 0.07 a | 0.39 $\pm$ 0.01 a | 5638.33 $\pm$ 262.76 a | 0.18 $\pm$ 0.01 a |
| M | 25 | 1786.3 $\pm$ 27.19 a | 241.38 $\pm$ 7.32 b | 2.66 $\pm$ 0.10 b | 0.49 $\pm$ 0.01 b | 9496.33 $\pm$ 237.45 b | 0.28 $\pm$ 0.01 a |
| I | | 1900.97 $\pm$ 46.07 a | 270.76 $\pm$ 6.43 a | 3.16 $\pm$ 0.09 a | 0.54 $\pm$ 0.01 a | 10425.67 $\pm$ 59.53 a | 0.31 $\pm$ 0.01 a |
| M | 35 | 2190.48 $\pm$ 31.17 b | 315.39 $\pm$ 18.34 b | 3.51 $\pm$ 0.15 b | 0.55 $\pm$ 0.01 a | 13428.33 $\pm$ 252.2 b | 0.54 $\pm$ 0.02 b |
| I | | 2451.63 $\pm$ 53.57 a | 390.05 $\pm$ 15.44 a | 5.09 $\pm$ 0.42 a | 0.60 $\pm$ 0.01 a | 14295.00 $\pm$ 117.41 a | 0.60 $\pm$ 0.01 a |

All values are presented as mean $\pm$ SE. The different letters in the same index and the same day indicate significant differences between treatments of mean values ($p < 0.05$, $t$-test).

The growth of watermelon plant analysis showed no significant difference between the monocropping and the intercropping systems at 15 and 25 day after transplanting. However, watermelon plant fresh and dry weights were significantly lower in the monocropping system than the intercropping system on day 35 after transplanting (Figure 1).

### 3.3. Effect of Wheat Intercropping on Incidence Rate of Fusarium Wilt of Watermelon

The Fusarium wilt symptoms appeared in watermelon plants in the monocropping system 15 days after transplanting. At 21 days after transplanting, some watermelon plants showed symptoms of Fusarium wilt in the intercropping system (Figure 2A,B). The occurrence rate of Fusarium wilt was higher (38 and 56%) in the monocropping system compared with the intercropping system (22 and 40%) at 25 and 35 days after transplanting, respectively (Figure 2A,C).

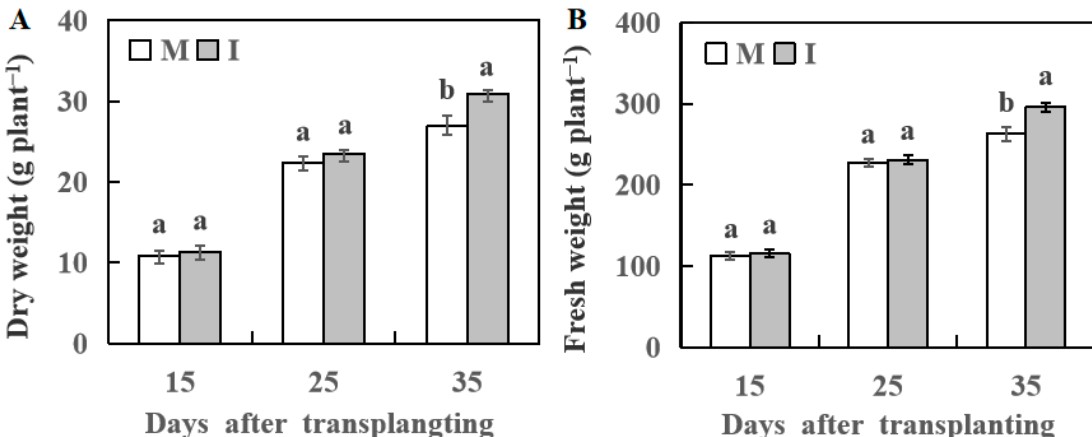

**Figure 1.** Dry weight of watermelon (**A**) and fresh weight of watermelon (**B**) at 15, 25, and 35 days after transplanting, respectively. M, watermelon plants grown in monoculture system; I, watermelon with wheat grown in the intercropping system. All values are mean ± SE (n = 3). The different letters on the columns of the same day indicate statistically significant differences between treatments ($p < 0.05$, *t*-test).

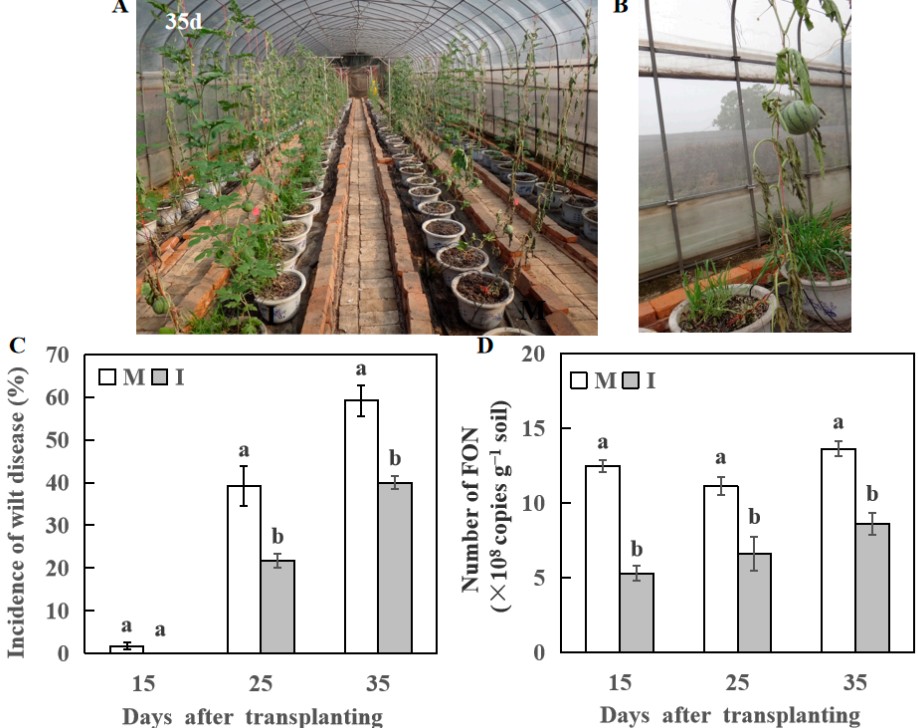

**Figure 2.** Changes in wilt disease of watermelon at 15, 25, and 35 days after transplanting, respectively. (**A**) Changes in phenotypes of watermelon at 35 day after transplanting. (**B**) The symptoms of Fusarium wilt in watermelon. (**C**) The incidence rate of wilt disease in watermelon. (**D**) The number of FON in the rhizosphere of watermelon. M, Watermelon plants grown in monoculture system; I, Watermelon with wheat grown in an intercropping system. All values are mean ± SE (n = 3). The different letters on the columns of the same day indicate statistically significant differences between treatments ($p < 0.05$, *t*-test).

The wheat-based intercropped plants of watermelon also showed fewer FON genomic DNA copies compared with monocropping watermelon plants at 15, 25, and 35 days after transplanting, respectively. A higher FON population was spotted in the monocrop-

ping system (13.61 × $10^8$ copies g$^{-1}$ soil) 35 days after transplanting (Figure 2D). It was significantly higher than the intercropping system.

### 3.4. Effect of Wheat Intercropping on the Expression of Defensive Enzyme Genes and Defensin-Like Genes

To estimate the defensive response to FON in watermelon roots, we observed variations in the expression levels of six defensive genes (Figure 3). The expression analysis of genes revealed that the *ClCAT* expression was 1.47-fold higher 1 day after the FON inoculation in the intercropping system compared to the monoculture system. Like *ClCAT* relative expression, we observed higher *ClSOD* expression in reaction to FON infection after 1 day. The *ClSOD* relative expression was highest at 1 and 3 days after the FON infection in the intercropping and monocropping systems, respectively. The relative expression of *ClAPX* and *ClCHI* peaked 3 days after FON inoculation, which was lower in the monoculture than in the intercropping system. The expression of *ClPPO* decreased 1 day after FON inoculation. Then, it was increased at 3 and 5 days, and the highest level was observed at 5 days after FON inoculation in the intercropping system than in the monocropping system. The expression of *ClPOD* declined after FON inoculation in intercropping and monocropping, respectively.

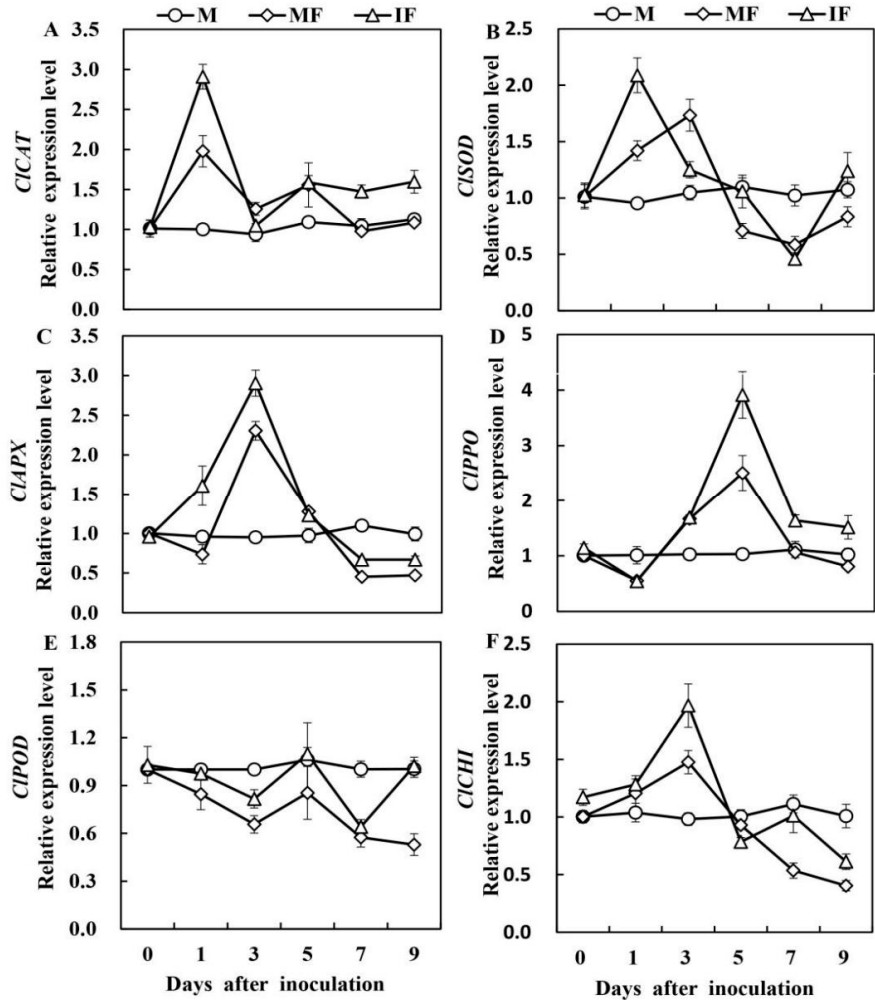

**Figure 3.** Relative expression levels of *ClCAT* (**A**), *ClSOD* (**B**), *ClAPX* (**C**), *ClPPO* (**D**), *ClPOD* (**E**), and *ClCHI* (**F**) defensive enzyme genes in watermelon roots in response to FON infection. M, Watermelon plants grown in a monoculture system; MF, Watermelon plants grown in a monoculture system with FON inoculation; IF, Watermelon with wheat grown in the intercropping system with FON inoculation. All values are mean ± SE (n = 3).

The defensin-like genes relative expression analysis revealed that *ClPDF2.1* and *ClPDF2.4* relative expression in the watermelon roots was increased after FON inoculation (Figure 4). The expression of *ClPDF2.1* was inclined 1 day after FON inoculation. At 3 days, expression in the intercropping was 1.2-fold higher than that of monocropping. In contrast, the higher relative expression of *ClPDF2.4* was quantified at 1 and 5 days after the FON inoculation. Relative expression of *ClPDF2.4* revealed a higher level of relative expression in the intercropping at 5 days after FON inoculation, which was 1.15-fold higher than that of the monoculture system.

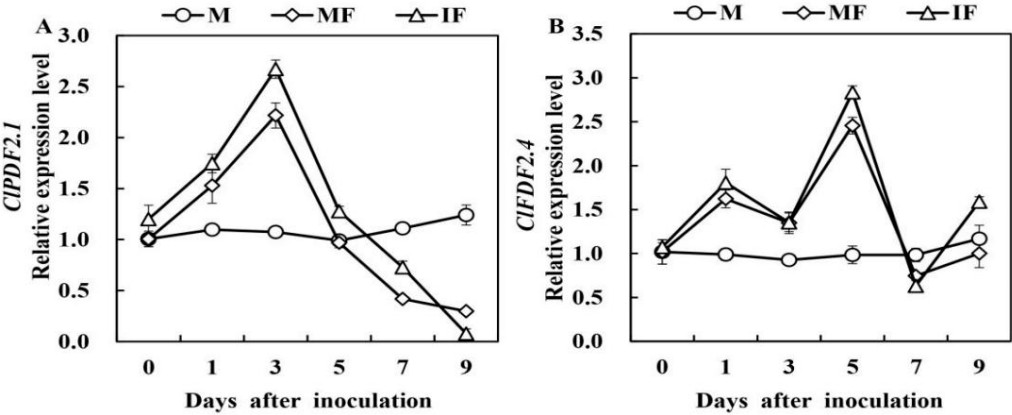

**Figure 4.** Relative expression levels of *ClPDF2.1* (**A**) and *ClPDF2.4* (**B**) defensin-like genes in watermelon roots in response to FON infection. M, Watermelon plants grown in a monoculture system; MF, Watermelon plants grown in a monoculture system with FON inoculation; IF, Watermelon with wheat grown in the intercropping system with FON inoculation. All values are mean ± SE (n = 3).

### 3.5. Effect of Wheat Intercropping on Defensive Enzyme Activities

The activity of the associated defensive enzymes (CAT, SOD, APX, and PPO) fluctuated at different inflection points. Still, it revealed similar trends, increasing first and then decreasing, except for POD (Figure 5). The CAT and SOD activities were enhanced by 13 and 19% in the intercropping system compared to the monocropping at 5 days after FON inoculation, respectively. Compared with the monocropping, the activity of APX was considerably increased in the intercropping at 3 days after FON inoculation and then declined after 7 days. The activity of PPO was higher in the watermelon roots of the intercropping than in the monocropping 1 day after FON inoculation. Finally, the POD activity was not different in the watermelon roots of the intercropping and monocropping systems. Furthermore, POD activity dropped below basal levels 3 days after being subjected to the FON inoculation.

### 3.6. Effect of Wheat Intercropping on Total Phenolic, Flavonoid, MDA and $H_2O_2$ Contents

Compared with the monocropping system, the total phenolic contents increased to 52 and 54% in the intercropping system at 1 and 9 days after FON inoculation (Figure 6A). The flavonoid contents were 20% higher in the intercropping than the monocropping 5 days after FON inoculation (Figure 6B).

The MDA contents gradually increased in watermelon roots for the intercropping and monocropping systems 1 day after FON inoculation. However, the increased level of MDA was notably lower in watermelon roots in the intercropping system than in the monocropping system (Figure 6C). Similarly, the $H_2O_2$ contents were significantly increased for both cropping systems after FON inoculation, peaked after 1 day, and then declined after 2 days. However, except for 1 day after FON inoculation, the $H_2O_2$ contents were reduced in the intercropping system than in the monoculture system (Figure 6D).

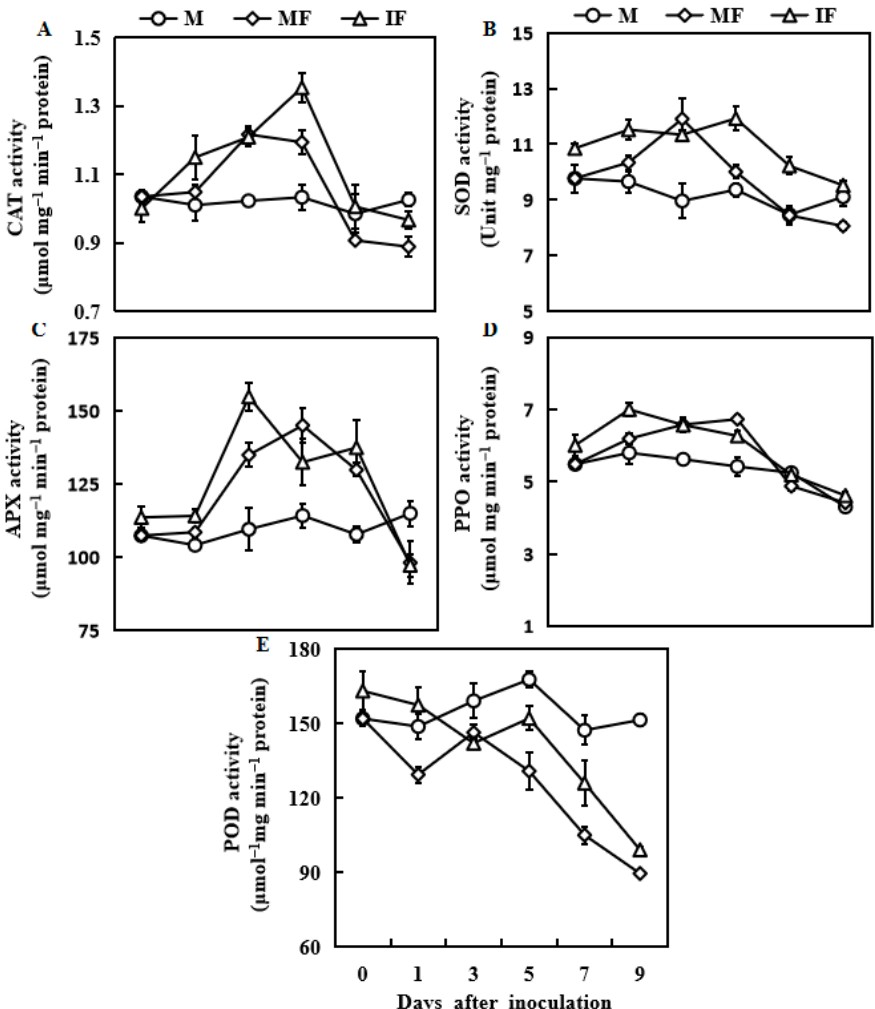

**Figure 5.** Responses of CAT activity (**A**), SOD activity (**B**), APX activity (**C**), PPO activity (**D**), and POD activity (**E**) in watermelon roots in response to FON infection. M, Watermelon plants grown in a monoculture system; MF, Watermelon plants grown in a monoculture system with FON inoculation; IF, Watermelon with wheat grown in the intercropping system with FON inoculation. All values are mean $\pm$ SE (n = 3).

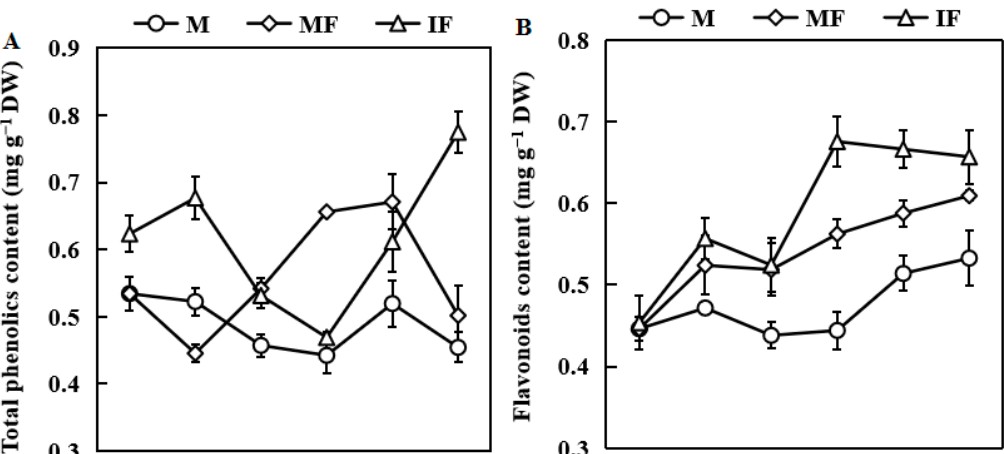

**Figure 6.** *Cont.*

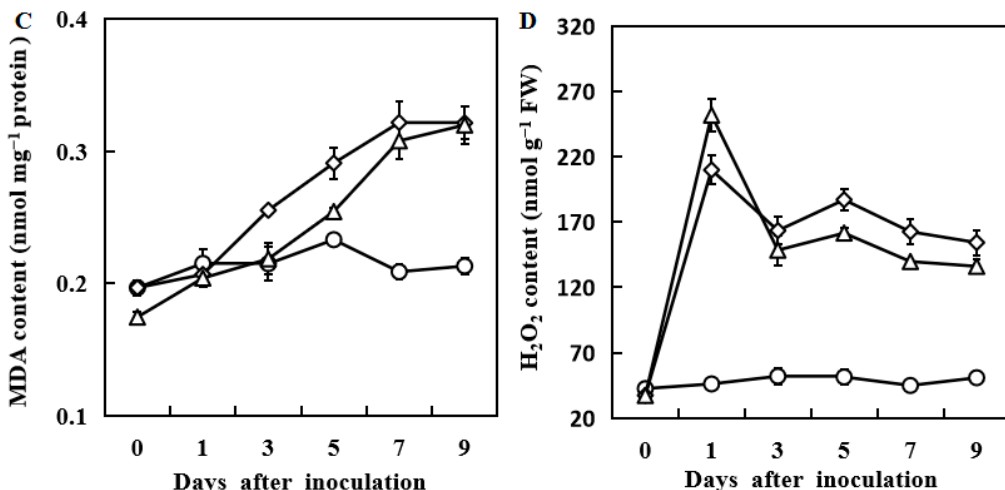

**Figure 6.** Responses of total phenolics content (**A**), Flavonoids content (**B**), MDA content (**C**), and $H_2O_2$ content (**D**), in watermelon roots in response to FON infection. M, Watermelon plants grown in a monoculture system; MF, watermelon plants grown in a monoculture system with FON inoculation; IF, Watermelon with wheat grown in the intercropping system with FON inoculation. All values are mean ± SE (n = 3).

## 4. Discussion

Fusarium wilting of watermelon is one of the serious soil-borne diseases that causes stunted growth and low production, resulting in important economic losses in many countries of the world [41]. Previous evidence demonstrated that intercropping (and companion cropping) systems could help plants against soil-borne diseases [1,3,42]. Again, Gao et al. [5] reported that root interaction between plant species in the intercropping systems was important for inhibiting pathogens and inducing disease resistance. Multiple studies have revealed that root exudates play an essential role in the defense against soil-borne pathogens [11,21,43]. Similarly, our results demonstrated that the population of FON and the occurrence of Fusarium wilt in watermelon reduced in the intercropping system after artificial inoculation with FON. Interestingly, we found that the mycelial growth of FON was inhibited by adding wheat root exudates, suggesting that wheat root exudates were directly involved in inhibiting the FON population in the rhizosphere and the occurrence of watermelon wilt.

Previous studies demonstrated that intercropping (and companion cropping) systems promoted plant growth [1–3,11]. Consistent with this, the results herein showed that the wheat-intercropped watermelon plants improved watermelon growth. Additionally, Wu et al. [44] found that intercropping with potato and onion promoted the growth of tomato seedlings. Together, these results indicated that wheat root exudates positively affect watermelon growth, which may also provide the benefit of decreased disease incidence.

Studies have explained that interspecific relations can improve plant resistance to disease by inducing disease-related defense-responsive gene expression [4,5]. Plans-induced defense responses were associated with many defense mechanisms, including activation of numerous defense-responsive genes, enzyme activity, and secondary metabolite synthesis [45,46]. The upregulation of defense-related genes, such as *ClPDF2.1*, *ClPDF2.4*, and *CHI*, plays an essential role in plant defense mechanisms against disease resistance, particularly in fungal pathogens [29,47]. Previously, CHI gene overexpression in tobacco enhanced transgenic plants' resistance to *Cercospora nicotianae* and *Rhizoctonia solani* [48]. Herein, the expression levels of *ClPDF2.1*, *ClPDF2.4*, and *ClCHI* genes were significantly enhanced in the wheat intercropping system, which indicates that FON and wheat root exudates might improve watermelon's capability to defend against disease progress.

The expression of *SOD*-encoding genes was significantly up-regulated when *Bemisia tabaci* was fed on ToCV-infected tomato for 24 h [49]. Suppression of APX resulted in hyper-

responsivity to pathogen infection was observed in transgenic tobacco plants [50]. Higher *PPO* levels in transgenic tomato plants make them more resistant to *Pseudomonas syringae* pv. tomato than wild plants [51]. In the current study, we also observed that FON inoculation enhanced expression levels of the *ClCAT*, *ClSOD*, *ClAPX*, and *ClPPO* genes in the intercropping system. However, the *ClPOD* gene expression level was downregulated in both cropping systems. By contrast, Li et al. [46] had recently reported that before and at days 1 and 3 after being subjected to *F. oxysporum* f. sp. n*iveum*, the *POD* gene expression increased in the wheat/watermelon intercropping than in the monoculture treatment. CAT, POD, PPO, and SOD are believed to be the active antioxidant enzymes for evaluating plants' physiological and biochemical responses to biotic and abiotic stresses [52]. Previous studies have reported that POD, PPO, and SOD activities are relatively higher in the cultivars capable of resisting Fusarium wilt in bananas after being treated with elicitor [53] or inoculated with *Fusarium oxysporum* f.sp. *cubense* [54]. Wu et al. [55] reported that the POD, PPO, and SOD activities in the roots of "Brazil Xiangjiao" were higher than controls 10 d after inoculation with Foc race 1. This study demonstrated that the CAT, SOD, APX, and PPO activities were also boosted in the watermelon roots in the intercropping after FON inoculation. However, the activity of POD dropped below basal levels after FON inoculation. With higher PPO activity, lignins and other phenolics were synthesized to strengthen cell walls, providing resistance against pathogen invasion [56,57]. Consistent with these, the results herein show that FON inoculation significantly triggered the expression of the primary disease-related defensive gene, which regulates the associated defensive enzyme activities in watermelon roots. Intriguingly, the timing and gene expression levels were more potent in the intercropping system than in the monoculture. Therefore, we assumed that the resistance of watermelon against FON was higher in the intercropping system due to the high expression of *ClPDF2.1*, *ClPDF2.4*, *CHI*, *ClCAT*, *ClSOD*, *ClAPX*, and *ClPPO* genes and due to the high activities of CAT, SOD, PPO, and APX defensive enzymes that prevented invasion from FON.

Total polyphenols and flavonoids in plants have redox properties by acting as reducing agents, hydrogen donors, and singlet oxygen quenchers [58,59]. In plants, phenolic compounds play the main role in wilt disease resistance by downregulating the enzymes (cellulase and pectin methyl esterase) associated with fungal cell wall degradation, thereby limiting cell wall degradation and fungal invasion [60]. Flavonoids are a group of extensive secondary metabolites previously demonstrated as promoters of disease resistance in different crops [61]. Herein, the total phenolic and flavonoid contents of watermelon roots were increased in the intercropping wheat system after FON inoculation. These results suggest that watermelon resistance to FON was enhanced in the intercropping wheat system because of an increasing accumulation of total phenolic and flavonoid contents in watermelon roots, which might improve watermelon's capability to protect against FON infection.

MDA content serves as an indicator of the degree of damage to plant cell membranes under stress. The higher the level of MDA, the more severe the damage to the cells [12]. This study showed that wheat intercropping reduced MDA content in watermelon roots after FON inoculation compared with a monoculture. It implied that wheat intercropping provided stability to biological membranes to resist the invasion of pathogens by decreasing MDA content. Interestingly, the $H_2O_2$ contents peaked 1 day after FON inoculation in the intercropping, and the $H_2O_2$ contents were lower in the intercropping than in the monoculture system. It is suggested that $H_2O_2$ can be decomposed into $H_2O$ and $O_2$ under the action of CAT in the intercropping system after FON infection [20,62], resulting in protecting the biological membrane from $H_2O_2$ injury, which plays an important role in preventing further invasion or expansion of FON. These results also showed that $H_2O_2$ as signaling seems to have strong signal transmission characteristics early in the intercropping system. The specific mechanisms of these effects require further analysis.

## 5. Conclusions

Intercropping with wheat improved watermelon growth and considerably reduced the disease occurrence of Fusarium wilt in watermelon. Moreover, the expression of disease-related defensin-like genes, defensive enzyme genes, defensive enzyme activities, and the total phenolic and flavonoid contents were induced more in watermelon plants in the wheat intercropping system than in watermelon plants grown in a monoculture system. However, compared with the monocropping system, the MDA contents in watermelon roots were reduced in the wheat intercropping system. The results suggested that intercropping with wheat alleviated the Fusarium wilt of watermelon by reducing the population of FON in rhizospheric soil, activating physiological responses and defense gene expression to protect watermelon from the pathogen, and improving the resistance of watermelon to FON in the watermelon with a wheat-grown intercropping system. Further research is needed to identify $H_2O_2$ as a signaling compound that has strong signal transmission characteristics at an early stage in the intercropping system.

**Author Contributions:** Conceptualization and methodology, Z.B. and H.L.; software, H.L. and J.L.; validation, Z.B. and Y.H.; formal analysis, H.L., J.L.; investigation and resources, Z.B., H.L., and M.W.; data curation, H.L. and J.L.; writing—original draft preparation, H.L and Z.B.; writing—review and editing, Z.B., H.L., Y.H., J.L., M.W., and C.Y.; visualization, J.L and Y.H.; supervision, project administration, and funding acquisition, Z.B., H.L., and C.Y. All authors have read and agreed to the published version of the manuscript.

**Funding:** This study was supported by China Agriculture Research System (CARS-25) to Z.B. and C.Y., by Scientific Research Startup Fund for High-level Talents, HeFei Normal University (2022rcjj02) to H.L., by Yong Talents Fund, Anhui Academy of Agricultural Sciences (QNYC-201906) to C.Y.

**Data Availability Statement:** The data presented in this study are available in the article.

**Conflicts of Interest:** The authors declare no conflict of interest.

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
