# Peer review of "Priming Watermelon Resistance by Activating Physiological Response and Defense Gene Expression to Alleviate Fusarium Wilt in Wheat-Watermelon Intercropping"

_horticulturae, doi:10.3390/horticulturae9010027_

Round 1

Reviewer 1 Report

he article "Priming watermelon resistance by activating physiological response and defense gene expression to alleviate Fusarium wilt in wheat-watermelon intercropping" has been reviewed for publication in Horticulturae. The authors present an interesting study on the effects of root exudates from the co-cultivation of cucurbits (melon) with cereals (wheat), where the positive effects in reducing the incidence of the pathogen (compared to monoculture) are characterized at the level of disease incidence, pathogenic inoculum potential, level of expression of genes related to plant defense or defensive enzymatic activities. The work is well planned and the experimental methodology seems to be adequate, although the English grammar and spelling should be reviewed (a series of suggestions and changes have been made directly to the manuscript in the attached pdf). In my opinion, the authors should have included an open field scale experimental design, where perhaps the beneficial effects of root exudates from co-cultivated wheat could be modulated by the presence of other microorganisms colonizing the rhizospheres of both plant hosts ( melon-wheat). 

Author Response

Respected Reviewer,

Thank you very much for your kind suggestions and changes to improve the quality of our article. Now we have carefully revised our article.

Reviewer 2 Report

The manuscript presents results scientifically valuable and important for horticulture practice. In Material and methods it would be advisable to provide more details in some places. In addition, there are a few typographical errors and unclear passages in the text. The lack of line numbering makes it difficult to add comments. After considering the comments, the manuscript should be published in Horticulture / MDPI

Remarks

- Throughout the text, the term 'Fusarium wilt' should be spelled normally, not italic, as it is understood as the name of the disease, not the proper name of the fungus

- chapter 1 line 2 it is Fusariu oxysporum, it should be Fusarium oxysporum

- page 2:

line 1-2  'and so on.’ should be removed

PAL - requires explanation

Trichoderma harzianum - it should be in italic

S.sclerotiorum - it should be Sclerotinia sclerotiorum (here it is used for the first time in the text - it must be without an abbreviation)

2.1

Fusarium oxysporum f. sp. niveum - please state how it was identified. In the present decade, molecular data should be submitted to the publicly available Genbank

- 'were carried out according to the methods described by [21] and [11].' A summary of what was done in this experiment should be given. It is hard for the reader to look for two other works. This passage is unclear in the current version.

 - 'was measured in two different directions.' Please specify what instrument was used to measure and with what accuracy (see Table 2 eg. 3.02 cm, 3.63 cm ... is such an accurate measurement possible since the hyphae do not grow perfectly evenly)

-Why was the evaluation carried out after 3 days, if FON does not grow fast, then e.g. after 6-7 days the differences would be more clear

- 'of 0.5 ml plant-1, 1 ml plant-1, 5 ml plant-1, and 10 ml plant-1.' What does  /plant mean, please provide more details (each Petri dish was taken with 1 ml of extracts).

- considerable suspension - whether only macroconidia or also microconidia were counted

-2.3 'methods described by Wu et al. [23]'. This methodology should be summarized. There are too many references to earlier papers, this makes it difficult to analyze the text

-Wu et al. [23],  it should be Wu et al. [23]

-2.5  what does 1   mean  'Cucurbit Genomics Database 1.'

Results

Table 2 - what does 'of 0.5 ml plant-1, 1 ml plant-1, 5 ml plant-1, and 10 ml plant-1’ mean

Figure 5 and Figure 6 it is advisable to write what does M, MF, IF (although it is stated in the methodology) mean

Figure 6 indicated to explain FW, DW

Discussion

 - In discussion should not be referred to Tables and Figures (only in exceptional circumstances). This needs to be corrected. Otherwise, the Discussion looks like Results.

- Cercosporanicotianae - should be Cercospora nicotianae

 - B.tabaci - it should be Bemisia tabaci (this is the name used for the first time in the text, so do not use abbreviations)

- F. oxysporum f. sp. Niveum, - it should be F. oxysporum f. sp. niveum,

- Conflflicts of Interest:    it should be Conflicts

- The authors declare no conflflict of interest - it should be conflicts ...

Literature:

The Latin spelling of proper names of plants and fungi should be standardized. They should all be spelled in italic

Literature 8     Eur.J. Soil. Biol.   needs correction

Literature 29   Plant. pathol. It should be Plant. Pathol.

Literature 46 it is 'Fusarium oxysporum f. sp. Niveum.' It should be Fusarium oxysporum f. sp. niveum

Literature 49  needs correction

Literature 57     Physiol. Mol. plant. P.  - needs correction

Author Response

Point 1: Throughout the text, the term 'Fusarium wilt' should be spelled normally, not italic, as it is understood as the name of the disease, not the proper name of the fungus

Response 1: Thank you for your suggestion. The term 'Fusarium wilt' of the entire manuscript has been revised.

Point 2: chapter 1 line 2 it is Fusariu oxysporum, it should be Fusarium oxysporum

Response 2: Thank you. We have correct the errors in the manuscript.

- page 2:

Point 3:line 1-2  'and so on.’ should be removed

Response 3: Thank you. 'and so on.’ in line 1-2 were deleted from the manuscript.

Point 4:PAL - requires explanation

Response 4: Thank you. PAL was deleted from the manuscript.

Point 5:Trichoderma harzianum - it should be in italic

Response 5: Thank you, we have revised in the manuscript.

Point 6:S.sclerotiorum - it should be Sclerotinia sclerotiorum (here it is used for the first time in the text - it must be without an abbreviation)

Response 6: Thank you. We have accepted and adopted“Sclerotinia sclerotiorum”.

2.1

Point 7:Fusarium oxysporum f. sp. niveum - please state how it was identified. In the present decade, molecular data should be submitted to the publicly available Genbank

Response 7: Thank you. Fusarium oxysporum f. sp. niveum was identified by referring to watermelon wilt disease-related literature: Zou, X.H. Identification and Inheritance of Resistance Analysis to Fusarium oxysporum f.sp. niveurm in Watermelon.2011; Geng, L.H.; Gong, G.Y.; Song, S.H.; Xu, X.L Wu, P.; Meng, S.C. Identification and control of seed-borne pathogen of watermelon wilt disease. J. Plant. Protection,2019,46: 330-336.

Point 8:-'were carried out according to the methods described by [21] and [11].' A summary of what was done in this experiment should be given. It is hard for the reader to look for two other works. This passage is unclear in the current version.

Response 8: Thank you. This section was rewritten as required in revised manuscript. Hopefully, it could be understood better.

Point 9:- 'was measured in two different directions.' Please specify what instrument was used to measure and with what accuracy (see Table 2 eg. 3.02 cm, 3.63 cm ... is such an accurate measurement possible since the hyphae do not grow perfectly evenly)

Response 9: Thank you. the diameter of colonies was measuredby referring to watermelon wilt disease-related literature:Ling, N.; Zhang, W.; Wang, D.; Mao, J.; Huang, Q.; Guo, S.; Shen,Q.R. Root exudates from grafted-root watermelon showed a certain contribution in inhibiting Fusarium oxysporum f.sp. niveum. PLoS One. 2013, 8, e63383.

Point 10:-Why was the evaluation carried out after 3 days, if FON does not grow fast, then e.g. after 6-7 days the differences would be more clear

Response 10: Thank you. By referring to FON mycelial growth-related literature, we found that FON mycelial growth were grow fast after 3 days in the related literature (Xu et al. 2015a).

Point 11:- 'of 0.5 ml plant-1, 1 ml plant-1, 5 ml plant-1, and 10 ml plant-1.' What does  /plant mean, please provide more details (each Petri dish was taken with 1 ml of extracts).

Response 11: Thank you. 0.5 ml plant-1 means 0.5 mL deionized water plant-1 for producing the root exudates. By referring to the root exudates concentration-related literature (Xu et al. 2015a). In addition, we also provided more details in the manuscript according to your suggestions.

Point 12:- considerable suspension - whether only macroconidia or also microconidia were counted

Response 12: Thank you. For suspension, only microconidia were counted.

Point 13:-2.3 'methods described by Wu et al. [23]'. This methodology should be summarized. There are too many references to earlier papers, this makes it difficult to analyze the text

Response 13: Thank you for your suggestion. This method was also added in material and methods section.

Point 14:-Wu et al. [23],  it should be Wu et al. [23]

Response 14:Thank you. We have revised the errors in the manuscript.

Point 15:-2.5  what does 1 mean 'Cucurbit Genomics Database 1.'

Response 15: Thank you for your question. 'Cucurbit Genomics Database now have update to version 2, when design the experiment only have version 1.

Results

Point 16:Table 2 - what does 'of 0.5 ml plant-1, 1 ml plant-1, 5 ml plant-1, and 10 ml plant-1’ mean

Response 16: Thank you. 0.5 ml plant-1 means 0.5 mLdeionized water plant-1for producing the root exudates. By referring to the root exudates concentration-related literature (Xu et al. 2015a).

Point 17:Figure 5 and Figure 6 it is advisable to write what does M, MF, IF (although it is stated in the methodology) mean

Response 17: Thank you for suggesting. We have added M, MF, and IF in Figure 5 and Figure 6.

Point 18:Figure 6 indicated to explain FW, DW

Response 18: Thank you. FW indicated fresh weight; DW indicated dry weight.

Discussion

Point 19: - In discussion should not be referred to Tables and Figures (only in exceptional circumstances). This needs to be corrected. Otherwise, the Discussion looks like Results.

Response 19: Thank you, tables and Figures were deleted in the manuscript.

Point 20:- Cercosporanicotianae - should be Cercospora nicotianae

Response 20: Thank you, we have accepted and modified in the manuscript.

 Point 21:- B.tabaci - it should be Bemisia tabaci (this is the name used for the first time in the text, so do not use abbreviations)

Response 21: Thank you, we have accepted and modified in the manuscript.

Point 22:- F. oxysporum f. sp. Niveum, - it should be F. oxysporum f. sp. niveum,

Response 22: Thank you, we have accepted and modified in the manuscript.

Point 23:- Conflflicts of Interest: it should be Conflicts

Response 23: Thank you, we have accepted and modified in the manuscript.

Point 24:- The authors declare no conflflict of interest - it should be conflicts ...

Response 24: Thank you, we have accepted and modified in the manuscript.

Literature:

Point 25:The Latin spelling of proper names of plants and fungi should be standardized. They should all be spelled in italic

Response 25: Thank you, we have accepted and modified in the manuscript.

Point 26:Literature 8     Eur.J. Soil. Biol.   needs correction

Response 26: Thank you, we have accepted and modified in the manuscript.

Point 27:Literature 29   Plant. pathol. It should be Plant. Pathol.

Response 27: Thank you, we have accepted and modified in the manuscript.

Point 28:Literature 46 it is 'Fusarium oxysporum f. sp. Niveum.' It should be Fusarium oxysporum f. sp. niveum

Response 28: Thank you, we have accepted and modified in the manuscript.

Point 29:Literature 49  needs correction

Response 29: Thank you, we have accepted and modified in the manuscript.

Point 30:Literature 57  Physiol. Mol. plant. P.  - needs correction

Response 30: Thank you, we have accepted and modified in the manuscript.

Thank you for your hard work again.

Reviewer 3 Report

In the manuscript entitled “Priming watermelon resistance by activating physiological response and defense gene expression to alleviate Fusarium wilt in wheat-watermelon intercropping”, the authors showed that intercropping wheat with watermelon offers watermelon some protection against Fusarium wilting. Fusarium oxysporum is a soil-borne disease that causes wilting in economically important crop such as watermelon. Several studies demonstrated that intercropping could alleviate the devastating effects of soil-borne diseases against economically crops.  In this study, the authors examined the effects of wheat root exudates from different cultivars on the growth of Fusarium mycelial growth. The results showed that the exudates inhibited Fusarium mycelial growth and watermelon-wheat intercropping reduced Fusarium wilt of watermelon. Expression of several defensive and defensin-like genes were also significantly induced in intercropping after inoculation with Fusarium.

The authors’ results support their conclusions. The presentation and analysis of the data are adequate for the overall objective. However, there are many grammatical issues in the manuscript.

Below are some of the sentences that should be revised.

Line 35 -38 – consider restructuring this section

“Watermelon is an important crop that is around the world. However, this plant is usually subjected to attack by Fusariu oxysporum f. sp. niveum (FON) in continuously cropped soils, which causes Fusarium wilt of watermelon. FON is one kind of soil-borne pathogen that is difficult to control due to the selectivity of host and its ability to survive in the soil for long periods of time.”

Line 101 -102 – consider revising this sentence

“In the second experiment, the experiment consisted of the following two treatments: (1) watermelon monoculture (M) and (2) watermelon with E18 wheat intercropping (I).”

Line 109 -111 - consider revising this sentence

“In the third experiment, the experiment consisted 109 of three treatments: (1) watermelon monocropping (M); (2) watermelon monocropping with FON inoculation (MF); (3) watermelon/wheat intercropping with FON inoculation 111 (IF).”

Line 116 -117 consider revising this sentence

“Root samples were harvested from watermelon plants after FON inoculation at points time points, frozen immediately in liquid nitrogen, and stored at 117 −80°C.”

Line 121 -125 consider restructuring this section

“The six whole plants per replicate were harvested at 15, 25 and 35 day after transplanting. The soil particles were removed from the roots with tap water, and the washed watermelon roots were scanned and analyzed root length, root surface area, root volume, root mean diameter, number of roots by root analyzer (EPSON Expression 11000XL). Drying were finished in oven. The dry weight of each plant was consisted of shoot dry weight and root dry weight. An electric balance was used to measure the dry weight of the plants.”

Line 131 consider revising this sentence

“The DNA was extracted from rhizosphere soil by PowerSoil DAN isolation kit (Omega Bio-Tek, Inc., GA, USA).”

Line 152 consider revising this sentence

“The defensive enzyme from root tissues were extracted following the procedure of 152 Cheng et al. [27].”

Line 191 consider revising this sentence

“However, E18 wheat root exudates showed potential against FON and the maximum inhibition rate was 10 % at 0.5 ml plant (Table 2).”

Line 206-207 consider revising this sentence

“The growth of watermelon plants were no significant difference between the mono-206 cropping system and the intercropping system at 15 and 25 day after transplanting.”

Line 226-227

“At 21 day after transplanting, some of the watermelon plants were began to develop wilt disease symptoms in the intercropping system.”

Author Response

Response to reviewers3Comments

Point 1: Line 35-38 – consider restructuring this section

“Watermelon is an important crop that is around the world. However, this plant is usually subjected to attack by Fusariu oxysporum f. sp. niveum (FON) in continuously cropped soils, which causes Fusarium wilt of watermelon. FON is one kind of soil-borne pathogen that is difficult to control due to the selectivity of host and its ability to survive in the soil for long periods of time.”

Response 1: Thank you. This sentence was revised in revised manuscript.

Watermelon is an important fruit and commercially cultivated. Fusarium wilt is one of the most severe soil-borne diseases and caused by Fusariu oxysporum f. sp. niveum (FON).

Point 2: Line 101 -102 – consider revising this sentence

“In the second experiment, the experiment consisted of the following two treatments: (1) watermelon monoculture (M) and (2) watermelon with E18 wheat intercropping (I).”

Response 2: Thank you. This sentence was revised in revised manuscript.

“In the second experiment, the assay included the following two croping systems. One is watermelon plants grown in monoculture system (M). The other one is watermelon with E18 wheat grown in intercropping system (I).”

Point 3: Line 109 -111 - consider revising this sentence

“In the third experiment, the assay included three treatments: (1) watermelon monocropping (M); (2) watermelon monocropping with FON inoculation (MF); (3) watermelon/wheat intercropping with FON inoculation (IF).”

Response 3: Thank you. This sentence was revised in revised manuscript.

“In the third experiment, the assay included three treatments: (1) watermelon plants grown in monoculture system (M); (2) watermelon plants grown in monoculture system with FON inoculation; (3) watermelon with E18 wheat grown in intercropping system with FON inoculation (IF).”

Point 4: Line 116 -117 consider revising this sentence

“Root samples were harvested from watermelon plants after FON inoculation at points time points, frozen immediately in liquid nitrogen, and stored at −80°C.”

Response 4: Thank you. This sentence was revised in revised manuscript.

Root samples harvested from monocropping and watermelon with wheat intercropping watermelon plants after FON inoculation at the indicated times, frozen immediately in liquid nitrogen and stored at −80°C prior to analyses.

Point 5:

Line 121 -125 consider restructuring this section

“The six whole plants per replicate were harvested at 15, 25 and 35 day after transplanting. The soil particles were removed from the roots with tap water, and the washed watermelon roots were scanned and analyzed root length, root surface area, root volume, root mean diameter, number of roots by root analyzer (EPSON Expression 11000XL). Drying were finished in oven. The dry weight of each plant was consisted of shoot dry weight and root dry weight. An electric balance was used to measure the dry weight of the plants.”

Response 5: Thank you. This sentence was revised in revised manuscript.

Sameple were harvested at 15, 25 and 35 day after transplanting and used for measure the dry weight of plants and root morphology. Root scanning was performed by using Imagery Scan Screen (Epson Expression 11000XL, Regent Instruments, Canada) and analyzed through WinRHIZO 2003a software (Regent Instruments, Canada). The samples were placed in an oven at 85℃ for drying and then weighted.

Point 6:Line 131 consider revising this sentence

“The DNA was extracted from rhizosphere soil by PowerSoil DAN isolation kit (Omega Bio-Tek, Inc., GA, USA).”

Response 6: Thank you. This sentence was revised in revised manuscript.

The DNA was extracted from rhizospheres soil by a PowerSoil DNA isolation kit (Omega Bio-Tek, Inc., GA, USA).

Point 7:Line 152 consider revising this sentence

“The defensive enzyme from root tissues were extracted following the procedure of  Cheng et al. [27].”

Response 7: Thank you, we have revised this sentence in the manuscript.

The defensive enzyme was measured as previously described [27].

Point 8:Line 191 consider revising this sentence

“However, E18 wheat root exudates showed potential against FON and the maximum inhibition rate was 10 % at 0.5 ml plant-1 (Table 2).”

Response 8: Thank you, we have revised this sentence in the manuscript.

E18 wheat root exudates have a greater inhibitory effect compared with E352 and Z9023, and the maximum inhibition rate was 9 % at 5 ml plant-1 (Table 2).”

Point 9:Line 206-207 consider revising this sentence

“The growth of watermelon plants were no significant difference between the mono-cropping system and the intercropping system at 15 and 25 day after transplanting.”

Response 9: Thank you. We have revised this sentence in revised manuscript.

The growth of watermelon plants analysis showed that no significant difference between the monocropping system and the intercropping system at 15 and 25 day after transplanting.

Point 10:Line 226-227

“At 21 day after transplanting, some of the watermelon plants were began to develop  symptoms in the intercropping system.”

Response 10: Thank you. We have revised this sentence in revised manuscript.

At 21 day after transplanting, some of the watermelon plants showed symptoms of Fusarium wilt in the intercropping system.

Thank you for your hard work again.
